# Clinical and Radiographic Outcomes and Treatment Algorithm for Septic Arthritis in Children

**DOI:** 10.3390/jpm13071097

**Published:** 2023-07-04

**Authors:** Alessia Caldaci, Gianluca Testa, Marco Simone Vaccalluzzo, Andrea Vescio, Ignazio Prestianni, Ludovico Lucenti, Claudia de Cristo, Marco Sapienza, Vito Pavone

**Affiliations:** Department of General Surgery and Medical Surgical Specialties, Section of Orthopaedics, A.O.U. Policlinico Rodolico-San Marco, University of Catania, 95123 Catania, Italy; alessia.c.92@hotmail.it (A.C.); gianpavel@hotmail.com (G.T.); marcovaccalluzzo@hotmail.it (M.S.V.); andreavescio88@gmail.com (A.V.); ignazioprestianni93@gmail.com (I.P.); ludovico.lucenti@gmail.com (L.L.); decristo.claudia@gmail.com (C.d.C.); marcosapienza09@yahoo.it (M.S.)

**Keywords:** arthrocentesis, arthroscopy, arthrotomy, hip, knee, pediatric, septic arthritis

## Abstract

Background: Septic arthritis (SA) in children is an acute inflammatory disease of the joints. If not treated promptly, it could become a surgical emergency. The incidence of the disease in children in Europe is approximately 2–7 per 100,000 children. The aim of this systematic review was to investigate which of these treatments—arthrocentesis, arthrotomy, and arthroscopy—provides better results in children and when to use them. Methods: Three independent authors conducted a systematic review of PubMed, ScienceDirect, and MEDLINE databases to assess studies with any level of evidence that reported the surgical outcome of SA. Two senior investigators evaluated and approved each stage’s findings. Results: A total of 488 articles were found. After screening, we chose 24 articles that were suitable for full-text reading based on the inclusion and exclusion criteria. The results of our analysis showed that there are no numerically significant differences reported in the literature on clinical and radiographic outcomes by surgical technique. Conclusions: We developed an algorithm that could be used if septic arthritis is suspected. Based on our results, the surgical technique to be used will depend on the operator who will perform it.

## 1. Introduction

Septic arthritis (SA) in children is an acute inflammatory disease of the joints that could become a surgical emergency [1]. The incidence in Europe is approximately 2–7 per 100,000 children [2]. SA is prevalent in children who are younger than 4 years old [3], especially in males more than females [4]. Different joints may be affected by the condition, but in 80% of cases, the hip and knee joints are involved [5].

From an etiological point of view, we can classify septic arthritis as primary or secondary. Primary septic arthritis is more common and is induced by direct injection of the pathogen or hematogenous dissemination. Despite *S. aureus* being the most common pathogen [6,7], we can distinguish other microorganisms based on the age of the children [8] (Table 1). *Kingella kingae* is a significant cause of septic arthritis in children under 4 years of age [9,10]. Secondary septic arthritis occurs when there is an infection that spreads to the joint. Most frequently, this happens when osteomyelitis in the intra-articular metaphyseal bone penetrates the joint cortex and seeds it [11].

The majority of children are healthy and free of any predisposing disorders [8,12]. The primary risk factors include: young age, male gender, illnesses such as hemoglobinopathies, host phagocytic abnormalities, and respiratory distress syndrome [8,12]. Additional risk factors include catheterization of the umbilical artery, urinary tract implants, and procedures of the urinary or digestive tract [8,12].

**Table 1 jpm-13-01097-t001:** The following table shows the main microorganisms responsible for pediatric septic arthritis, divided by age. Staphylococcus aureus is the most prevalent pathogen in all age groups [8].

Age	Most Common Microorganisms
<3 months	Staphylococcus aureus; Streptococcus agalactiae; Neisseria gonorrhoeae; Candida;
3 months–5 years	Staphylococcus aureus; Streptococcus; Kingella kingae; Streptococcus pneumoniae; Haemophilus influenzae type b;
>5 years	Staphylococcus aureus; Streptococcus; Streptococcus pneumoniae; Salmonella; Neisseria meningitidis.

The classic clinical picture of septic arthritis in children is a painful joint with limited movement coupled with fever and malaise [13]. Flexion, abduction, and external rotation of the leg are typical symptoms of a pediatric hip infection [14].

Positive outcomes depend on an early diagnosis, which is based on the patient’s complete medical history, physical examination, imaging, laboratory tests, and aspiration of the suspicious joint [11].

For the diagnosis, it is useful to evaluate Kocher’s criteria, as described in 1999 [15], highlighting four specific criteria that could be used to differentiate septic arthritis from transient synovitis. Positive “Kocher Criteria” were listed as follows: white blood cell count (WBC) ≥ 12,000/µL, erythrocyte sedimentation rate (ESR) ≥ 40 mm/h, temperature ≥ 38.51 °C, inability to bear weight, and C-reactive protein (CRP) ≥ 20 mg/L [14].

To prevent complications caused by increased intra-articular pressure and decreased perfusion, which could lead to osteonecrosis, it is critical to start therapy within the first five days following the development of symptoms. The first approach consists of starting empirical antibiotic therapy with new-generation cephalosporins, vancomycin, or clindamycin in suspected MRSA infection, followed by a specific therapy for the pathogen with a total duration of 2 to 6 weeks, switching from intravenous to oral therapy upon normalization of CRP and WBC count [16]. Once the antibiotic therapy has started, there are three possible types of orthopedic treatment: arthrocentesis, arthrotomy, and arthroscopy. Arthrocentesis consists of a sampling, by aspiration, of synovial fluid contained in the relative joint capsule [17]. It should be performed within 5 days of the onset of symptoms for best results. It could be echo-guided [18], and the patient may be sedated or under local anesthesia, performing joint irrigation with saline solution when the aspirated contents are particularly purulent [2]. Arthrotomy is a surgical procedure that examines a joint, looking at the ligaments, cartilage, and intra-articular structures [12]. Reduced bacterial load, necrotic tissue removal, joint decompression, and, if necessary, drilling of the metaphysis in situations with concomitant osteomyelitis are possible with open debridement [19]. Arthroscopy is a minimally invasive surgical procedure. Through a small incision, a surgeon inserts a skinny tube connected to a fiber-optic video camera. A high-definition video display receives the view inside the joint [20,21,22]. The advantage over traditional open surgery is that the joint does not have to be opened up fully [20].

Nowadays, the best joint drainage procedure is still debated: arthroscopy, arthrotomy, or arthrocentesis. The aim of this systematic review was to investigate the indications and outcomes of these procedures for the main affected joints, the hip and knee, providing a diagnostic-therapeutic guide for the orthopedic surgeon.

## 2. Materials and Methods

### 2.1. Search Selection

A PIOS approach was utilized to carry out the study: Patient (P); Intervention (I); Outcome (O); and Study Design (S). The literature assessing treatment (I) in septic arthritis-affected patients (P) was analyzed. According to clinical and radiographic outcomes, complications have been evaluated based on accuracy (O). The following study designs were included (S): Randomized Controlled Trials (RCT), Prospective (PS), Retrospective (RS), Case series (CS), Case-Control (CC), and Cohort (C) studies. The data currently available in the literature on the outcome of treatment of septic arthritis disease were analyzed overall. The article selection process was conducted in accordance with the Preferred Reporting Items for Systematic Review and Meta-Analyses (PRISMA) guidelines. The Study was conducted from May to September 2022 using Scopus, Web of Science, PubMed, Science Direct, and MEDLINE databases. The search string included the terms “arthrocentesis”, “arthroscopy”, “arthrotomy”, “hip”, “knee”, “pediatric”, and “septic arthritis”. Three different operators independently analyzed the texts (A.C., I.P. and M.S.V.). Two senior surgeons (V.P. and G.T.) consultations were used to resolve disputes (Table 2).

### 2.2. Study Selection

After research, 488 articles were found. The analysis of the texts, conducted according to the inclusion and exclusion criteria, selected 24 eligible articles for the final analysis. The process of selection and screening is outlined in the PRISMA flowchart. (Figure 1).

The inclusion criteria for the eligible articles were: (1) a known affected joint (knee or hip), (2) age < 18 years, (3) a definite diagnosis of septic arthritis, (4) a complete searchable article, (5) a specified treatment used (arthrocentesis, arthrotomy, or arthroscopy), and (6) articles in English.

The exclusion criteria were: (1) patients with osteomyelitis that could invalidate outcomes and (2) treatment used that was unclear.

### 2.3. Data Extraction

Three reviewers (A.C., M.S.V. and I.P.) independently analyzed the articles. The extracted data were number and location of affected joints, mean age, average symptoms duration time, mean follow-up, type of treatment, percentage of re-intervention, clinical, and outcomes.

### 2.4. Outcome Measures

Patients were evaluated based on clinical outcome, ROMs, residual pain evaluation, and radiographic assessment, consisting of the presence of residual bone alterations and the need to resort to additional treatments.

## 3. Results

### 3.1. Demographics

Treatments were given to 395 hips and 177 knees, with a mean age of onset of 6.3 years (range 0–12 years), an average duration of symptoms of 4.5 days, and a mean follow-up of 6 years.

### 3.2. Treatment of Septic Arthritis

Six articles looked at hip arthrocentesis [23,24,25,26,27,28], six articles evaluated hip arthrotomy [29,30,31,32,33,34], five articles examined hip arthroscopy [24,35,36,37,38], five articles treated knee arthrocentesis [34,39,40,41,42], three articles analyzed knee arthrotomy [34,43,44], and three articles described knee arthroscopy [44,45,46].

### 3.3. Type of Initial Treatment for Septic Hip Arthritis

Of the 395 examined, 47% received arthrocentesis treatment, 12% arthroscopic treatment, and 41% arthrotomic treatment (Figure 2).

### 3.4. Percentage of Reoperation in Septic Hip Arthritis by Type of Initial Treatment

Of the 47% of patients who received initial treatment with arthrocentesis, 70% required a new evacuative arthrocentesis and 15% a resolving arthrotomy. Of the 12% of patients treated with arthroscopy, 15% required a second resolutive arthroscopy procedure. Finally, of the 41% of patients treated with arthrotomies, 3% needed a second evacuative arthrotomy (Figure 3). On average, reoperation occurred within 14 days of the first procedure.

### 3.5. Percentage of Clinical Complications in Septic Hip Arthritis by Type of Initial Treatment

Only 4% of individuals who had arthrocentesis initially experienced clinical consequences, including ROM restriction and ongoing discomfort; 6% in patients treated with arthrotomy; and no cases in patients treated with arthroscopy (Figure 4a).

### 3.6. Percentage of Radiological Complications in Septic Hip Arthritis by Type of Initial Treatment

Regarding radiological results, 2% of patients who underwent arthroscopy, 6% of patients who underwent arthrocentesis, and 8% of patients who underwent arthrotomy showed at medium-term follow-up, about 5 years, radiological outcomes such as coxa magna, ossification nucleus smaller than contralateral, high metaphysical bone reactions, heterotopic ossification, and avascular necrosis of the head (Figure 4b).

### 3.7. Type of Initial Treatment for Septic Knee Arthritis

Of the 177 knees also examined, 60% received arthrocentesis treatment, 26% received arthroscopic treatment, and 14% received arthrotomic treatment (Figure 5).

### 3.8. Percentage of Reoperation in Septic Knee Arthritis by Type of Initial Treatment

Of the 60% of patients who received initial treatment with arthrocentesis, 22% required a new evacuative arthrocentesis, 7% a resolving arthrotomy, and 1% arthroscopy surgery. Of the 26% of patients treated with arthroscopy, 2% required a second resolutive arthroscopy procedure, and 2% required a new evacuative arthrocentesis. Finally, of the 14% of patients treated with arthrotomies, 10% needed a second arthrotomy (Figure 6).

### 3.9. Percentage of Clinical Complications in Septic Knee Arthritis by Type of Initial Treatment

Overall, only 7% of patients initially treated with arthrocentesis developed clinical complications such as ROM limitation and persistent pain, 4% in patients treated with arthroscopy, and no cases in patients treated with arthrotomy (Figure 7a).

### 3.10. Percentage of Radiological Complications in Septic Knee Arthritis by Type of Initial Treatment

Regarding radiological outcomes, 8% of patients treated with arthrocentesis and 6% of patients treated with arthroscopy showed at medium-term follow-up, about 5 years, radiological outcomes such as tibial or femoral bone necrosis and alterations of the articular rim (Figure 7b).

## 4. Discussion

As septic arthritis in children is a disease that can become a medical emergency, treatment should be prompt and appropriate. Unfortunately, to date, there are no official guidelines available on which surgical technique to prefer between arthroscopy, arthrotomy, or arthrocentesis. Each of these surgical techniques has advantages over the others.

Direct joint visualization, the capacity to completely remove the necrotic synovium, and the ability to thoroughly irrigate the joint with less invasive surgery represent the benefits of arthroscopy, a quicker and less invasive technique. In comparison to arthroscopy or arthrotomy in a very young child, this procedure may be simpler with the help of ultrasonography and, if necessary, anesthesia. An arthrotomy drawback is a larger incision with more scar tissue, which offers a better view of the joint and permits enough irrigation.

Our findings indicated that, compared to the other two methods, arthrotomy results in a lower risk of reoperation in hip septic arthritis. Only 3% of the patients treated with arthrotomies required a second resolutive surgery.

On the other hand, the study of Ross et al. [47] reported that hip arthroscopy reduces recovery duration compared to traditional open exposures. Sanpera et al. [35] focused on the effectiveness of arthroscopy for hip arthritis in children, reporting excellent results in 20 children arthroscopically treated; the study evidenced arthroscopy as a useful form of treatment before resorting to an arthrotomy.

Concerning knee septic arthritis, our systematic review underlined that the use of arthroscopy has a lower risk of reoperation. In fact, only 2% of patients who underwent arthroscopy needed a second resolutive surgery. Clinical and radiographic alterations following the use of these surgical techniques were evaluated. According to our research, 2–6% of patients who were being treated for septic arthritis of the hips had clinical signs, and 2–8% had radiographic alterations. Otherwise, in patients treated for septic arthritis of the knee, clinical changes were reported in 0–10% of patients and radiographic alteration in 0–15%. These data are in line with the study of Hoswell et al. [48], which evidenced that septic arthritis of the knee had a slightly higher percentage of good clinical and radiographic outcomes than septic arthritis of the hips.

Analyzing the management option, our data showed there are no numerically significant differences over the medium-long term in the clinical and radiographic outcomes for the operator’s treatment technique. There are no studies in the literature that really indicate the best treatment for septic arthritis in children. A systematic review for the optimal management of septic arthritis in children, conducted in 2009 by Kang et al. [49], did not provide any indication as to the type of treatment to be preferred. The authors proposed a multidisciplinary approach that involves surgeons, pediatricians, radiologists, microbiologists, and nurses without providing a solution to the dilemma.

In 2021, Donders et al. [13] reported that there may be fewer dangers associated with pediatric arthroscopy compared to arthrocentesis and arthrotomy. According to the authors, due to its minimally invasive nature, arthrocentesis of the knee may be advantageous in extremely young children.

Finally, in accordance with our data, El-Sayed et al. [37] conclude that immediate and complete joint septic drainage is the key to successful treatment, regardless of the technique choice.

The limitations of this study were the heterogeneity of the data analyzed and the highly different age groups in the various research studies, which may affect the outcomes.

## 5. Conclusions

The rate of clinical and radiological complications by technique is uniformly low; therefore, the choice is purely operator-dependent. Further studies will be necessary to create updated treatment guidelines.

Lastly, we finish by sharing our diagnostic algorithm, which we believe will be helpful if pediatric septic arthritis is suspected (Figure 8).

## Figures and Tables

**Figure 1 jpm-13-01097-f001:**
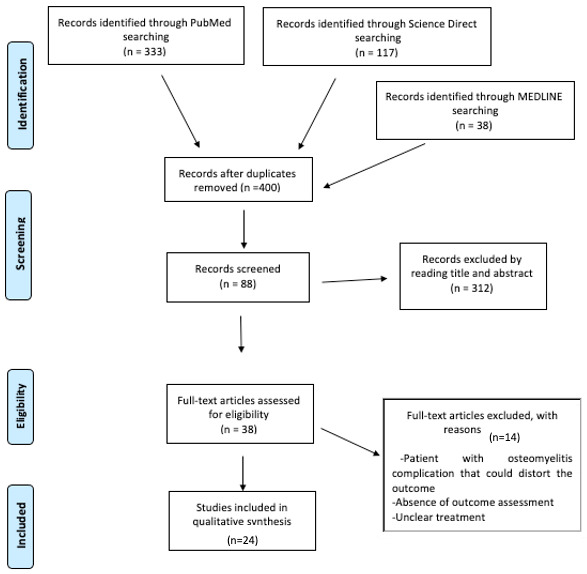
PRISMA (Preferred Reporting Items for Systematic Reviews and Meta-Analysis) flowchart of the systematic literature review.

**Figure 2 jpm-13-01097-f002:**
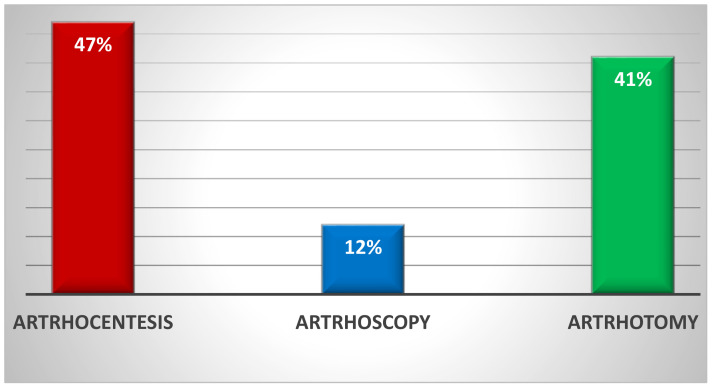
The figure above shows the type of initial treatment used for septic arthritis of the hip.

**Figure 3 jpm-13-01097-f003:**
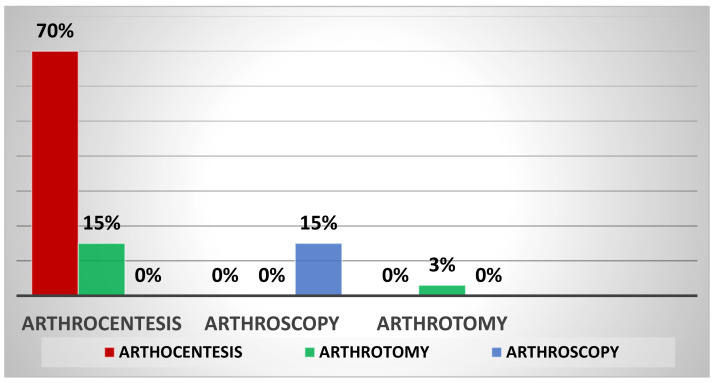
The figure above shows the reoperation rate in septic hip arthritis based on the type of initial surgical technique.

**Figure 4 jpm-13-01097-f004:**
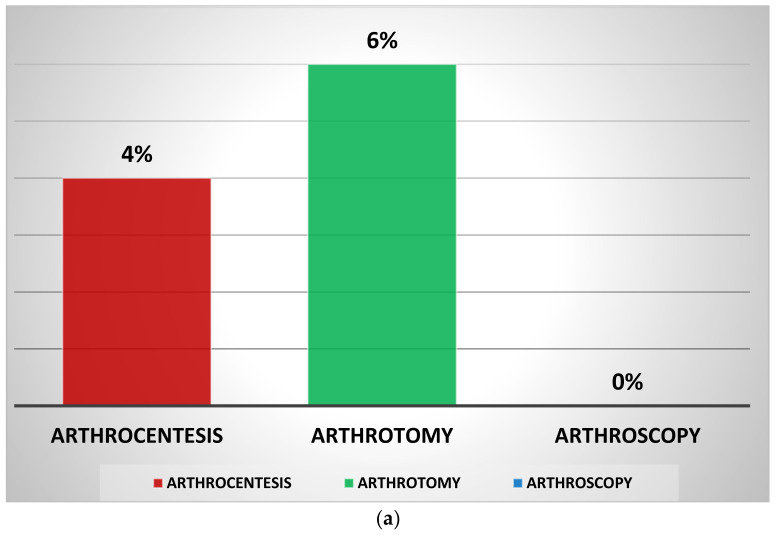
Treatment for septic hip arthritis: (**a**) percentage of clinical complications by initial technique and (**b**) percentage of radiological complications by initial technique.

**Figure 5 jpm-13-01097-f005:**
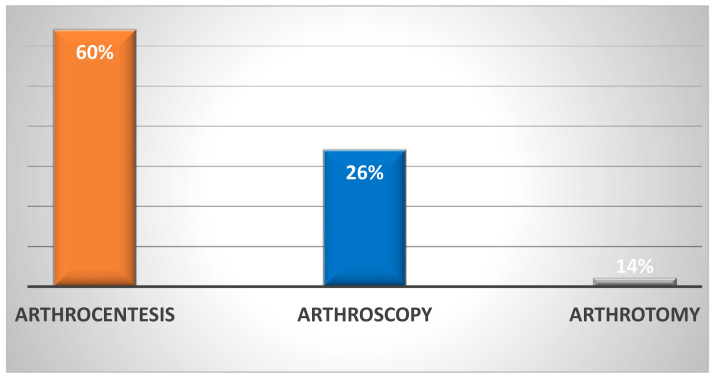
The figure above shows the type of initial treatment used for septic arthritis of the knee.

**Figure 6 jpm-13-01097-f006:**
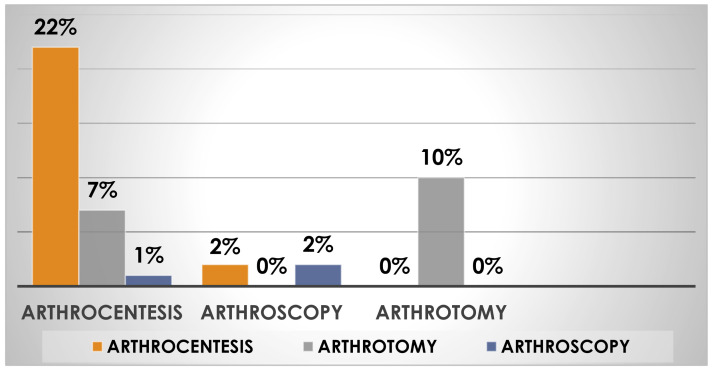
The figure above shows the reoperation rate in septic knee arthritis based on the type of initial surgical technique.

**Figure 7 jpm-13-01097-f007:**
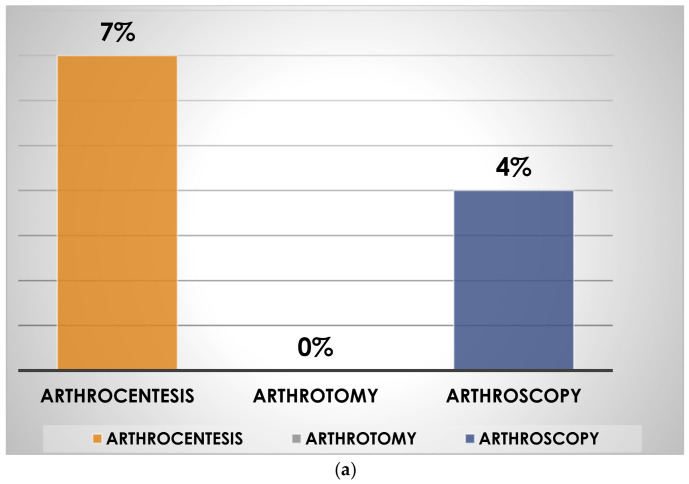
Treatment for septic knee arthritis: (**a**) percentage of clinical complications by initial technique and (**b**) percentage of radiological complications by initial technique.

**Figure 8 jpm-13-01097-f008:**
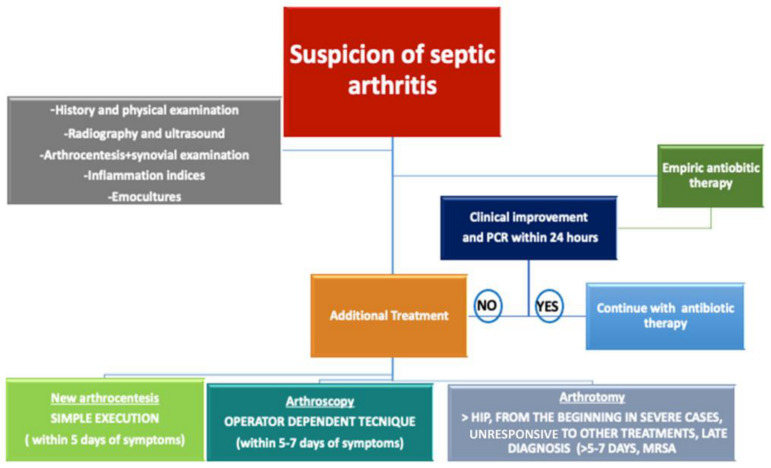
Treatment algorithm in cases of suspicion of septic arthritis: First step (follow the points in the gray rectangle at the top left); Second Step (green rectangle): perform empiric antibiotic therapy; and Third Step (blue rectangle): the patient must be re-evaluated after 24 h. If the patient has improved, continue with the antibiotic therapy. If the patient has not improved, surgery should be continued. The choice between arthrocentesis, arthrotomy, and arthroscopy depends on the skills of the surgeon and the time elapsed since the beginning of the symptom.

**Table 2 jpm-13-01097-t002:** Studies included in the systematic review.

Ref	Author	Joint	No. of Joints	Treatment	Mean Follow-Up	Radiological Outcome	Clinical Outcome
[23]	Kotlarsky P. et al. (2016)	Hip	14	AS	6–8 Y	None	None
[24]	Thomas M. et al. (2021)	Hip	103	AS + AT (82) AC (21)	Nk	Nk	Nk
[25]	Byani A. et al. (1988)	Hip	42	AS	1–3 Y	Coxa Magna, Joint destruction	Limitation of ROM
[26]	Givon U. et al. (2004)	Hip	28	AS	2–9 Y	None	None
[27]	Pääkkönen (2010)	Hip	45	AS	>1 Y	None	None
[28]	Griffet J. et al. (2011)	Hip	19	AS	1–3 Y	Smaller ossification nucleus, Coxa Magna	None
[29]	Lyon R.M. et al. (1999)	Hip	25	AT	4 M–7 Y	Heterotop ossification, Coxa-Magna	None
[30]	Samilson S.R. et al. (1958)	Hip	7	AS	10 Y	None	None
[31]	Bennett O.M. et al. (1992)	Hip	45	AT	2–5 Y	Coxa Magna, Ischemic necrosis of epiphysis, dysplasia of the acetabolum	Reduction of ROM, residual pain, limb length discrepancy
[32]	Umer et al. (2003)	Hip	40	AT	1–2 Y	Partial growth plate, partial avascular necrosis of the femoral epiphysis	Reduction of ROM
[33]	Kim H.M. et al. (2000)	Hip	20	AT	1–5 Y	Smaller ossification nucleus, Coxa Magna	Residual pain
[34]	Wiley J.J. et al. (1999)	Hip Knee	16 15 + 7	AT AC + AT	Nk Nk	Avascularal necrosis None	Reduction of ROM None
[35]	Sanpera I. et al. (2016)	Hip	11	AS	3–8 Y	Metaphyseal modification	None
[36]	Chung W.K. et al. (1993)	Hip	9	AS	Nk	Metaphyseal modification	None
[37]	El- Sayed et al. (2008)	Hip	10	AS	1–3 Y	None	None
[38]	Fernandez F. et al. 2013)	Hip	18	AS	6 M–4 Y	Avascular necrosis	Limitation of the ROM
[23]	Kotlarsky P. et al. (2016)	Knee	17	AC	4 Y	None	None
[39]	Halder D. et al. (1996)	Knee	9	AC	3–16 M	None	Limitation of the ROM
[40]	Strong M. et al. (1954)	Knee	50	AC	1–21 Y	None	Pain, limb length discrepancy, reduction of ROM
[41]	Herdon W.A. et al. (1986)	Knee	15	AC	1–5 Y	None	None
[42]	Katz K. et al. (1990)	Knee	5	AT	1–3 Y	Partial necrosis of the tibial plate	None
[43]	Johns B. et al. (2018)	Knee	13 11	AT AS	1–20 Y	None	None
[44]	Smith M.J. et al. (1956)	Knee	20	AS	6–60 M	None	None
[45]	Stanitski C.L. et al. (1989)	Knee	15	AS	30–48 M	None	None

AS = arthroscopy, AC = arthrocentesis, AT = arthrotomy, M = months, Y = years, and Nk = Not known.

## Data Availability

The data presented in this study are available in the article.

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
