# Peer review of "Clinical and Radiographic Outcomes and Treatment Algorithm for Septic Arthritis in Children"

_jpm, 2023, doi:10.3390/jpm13071097_

Round 1
Reviewer 1 Report
Thank you for the opportunity to review this paper.
Although the paper is a systematic review of septic arthritis in children, it is thought to have many revisions.
1. In general, the septic arthritis in children is surgical emergency. The most important treatment of septic arthritis in children is joint irrigation and drainage.
Why is the septic arthritis medical emergency? And why is the antibiotic therapy (second step) more advanced step than surgical step (third step)?
2. If the septic arthritis is diagnosed, the surgical treatment is performed as soon as possible because the most important prognostic factor is early diagnosis and early drainage.
In figure 8, why does the re-evaluation need to be performed after 24 hours?
3. In figure 8, what is the evidence of choice between arthrocentesis, arthroscopy and arthrotomy according to the symptom duration?
4. All of the figure, there was no statistical difference. The authors should perform statistical analysis.
5. The age is very important factor for choice of surgical treatment in septic arthritis in children. Although the authors mentioned the limitation, the analysis according to age should be performed.
6. In septic arthritis, the drainage of joint is very important. So the most of the surgeons insert the drain (penrose drain or JP drain). However, the drain cannot be inserted in arthrocentesis. I think this is the cause of high rate of reoperation in arthrocentesis. Moreover, the sedation or anesthesia may be required in reoperation. I think the arthrocentesis is not good choice of surgical treatment in septic arthritis. The authors should mention it.
7. There was no clinical outcome. the authors only analysis the clinical complication.
8. In figure legend of figure 5, the hip should be changed to knee.
none
Author Response
1)Thanks for the comment, I corrected medical with surgicaI
I Don't understand disappointment about antibiotic therapy. As specified in my text, it represents the step that precedes any form of drainage (once the antibiotic therapy has started, there are 3 possible different types of orthopaedic treatment: arthrocentesis, arthrotomy and arthroscopy)
2)We agree with his thought, in fact the diagnostic and therapeutic procedure of drainage with arthrocentesis has been inserted in the gray box. The 24H re-evaluation allows us to understand if the clinical and hematological conditions have improved, if not, we will proceed with a second drainage procedure (arthrocesis, arthroscopy or arthrotomy)
3)The evidence is scientific and the related studies can be found in a note in the introductory part relating to the description of the various drainage techniques
4)Thanks for the comment, I corrected it as not statistically significant, which could suggest a statistical analysis, with numerically insignificant differences. Being a review and not a meta-analysis, it must limit itself to reporting the data sterilely. (However, the percentages of clinical complications 6% and 8% and radiographic 4% and 6% are almost comparable to the eye)
5)The study was not conducted by age as even the few articles in the literature make little mention of it therefore the data collection would not have been conspicuous
6)Thanks for the comment, but I didn't want to associate the high rate of post-arthrocentesis recurrence with the lack of drainage at the site as the most accepted reason in the literature is given by the fact that arthrocentesis does not allow, unlike the other techniques, an empty full initial of the joint. It certainly won't be the best technique, but it is the first to be used not only for its therapeutic purpose, but also for diagnosis and in some centers the only technique that can be used in acute cases
7)By clinical outcome we mean the clinic (signs and symptoms) that can be objectified on the patient at follow-up (a term also used by the studies examined to differentiate from the radiographic component)
8)Sorry for the oversight, we have made the correction

Reviewer 2 Report
Overall, this is a well executed systemic review comparing surgical outcomes for septic arthritis in children. No major scientific flaws are apparent. Some minor comments for improvement include the following:
1. The abstract mentions only that "treatments" are being compared without specifying that the comparison is specifically focused on arthroscopy, arthrotomy, and arthrocentesis. This should be mentioned since it is essential to the study's design.
2. The treatment algorithm in Figure 8 is very colorful and able to be followed but could benefit from using standard symbols, e.g., https://en.wikipedia.org/wiki/Flowchart#Common_symbols
English language is understandable throughout, but the overall presentation could benefit from further proofreading. For example line 15: "in-vestigate" should be without a hyphen. Lines 14 and 30: "100.000" should be written with comma instead of period. Line 67: "eco-guided" is contains a misspelling of "echo", which should probably be replaced with the word ultrasound, resulting in "ultrasound-guided" as a more common term. Line 185: "ast" should be "as". In Figure 8, in the "ARTHROTOMY" box, the word "responsible" is misused (likely should be "responding"), and the closed parenthesis is missing at the end.
Author Response
1)Thanks for the valuable advice, I have corrected the text
2)Unfortunately we had to simplify the graph to a minimum as we had pagination and layout problems
3)Thanks for the careful advice, we have made the corrections

Round 2
Reviewer 1 Report
It appears that all comments and suggestions have been adequately answered.